# Timely Association of RSV Hospitalisation Waves in Children with the Detection of SARS-CoV-2 in the General Population in Eastern Bavaria

**DOI:** 10.3390/v17121584

**Published:** 2025-12-05

**Authors:** Alexander Kiefer, Andreas Ambrosch, Benedikt M. J. Lampl, Fritz Schneble, Kai Rubarth, Stefan Vlaho, Matthias Keller, Michael Kabesch

**Affiliations:** 1Department of Paediatric Pneumology and Allergy, University Children’s Hospital Regensburg (KUNO), Hospital St. Hedwig of the Order of St. John, University of Regensburg, 93049 Regensburg, Germany; 2Institute of Laboratory Medicine, Microbiology and Infection Prevention, Hospital of the Order of St. John, 93049 Regensburg, Germany; 3Division of Infection Control and Prevention, Regensburg Department of Public Health, 93059 Regensburg, Germany; benedikt.lampl@landratsamt-regensburg.de; 4Department of Epidemiology and Preventive Medicine, University of Regensburg, 93053 Regensburg, Germany; 5Department of Paediatrics, 92637 Weiden, Germany; 6Department of Paediatrics, Sana Klinikum Hof, 95032 Hof, Germany; kai.rubarth@sana.de; 7Department of Paediatrics, Innklinikum Altötting, 84503 Altöttiing, Germany; 8Kinderklinik Dritter Orden Passau, 94032 Passau, Germany; 9Science Development Campus Regensburg (WECARE), Hospital St. Hedwig of the Order of St. John, 93049 Regensburg, Germany

**Keywords:** RSV, SARS-CoV-2, hospitalisation, epidemiology

## Abstract

Background: The Respiratory Syncytial Virus (RSV) season of 2023/2024 was conspicuously different from previous seasons, with an abrupt decrease in hospitalisation rates at the peak of the season, leading to two lower peaks instead of the expected one high peak of hospitalisations. Thus, we investigated whether there was an interference with other virus infection waves in the course of that RSV season. Methods: We analysed RSV seasons since 2016 in children who were hospitalised due to an RSV infection in Eastern Bavaria and investigated epidemiological features of RSV seasons after the severe acute respiratory syndrome coronavirus 2 (SARS-CoV-2) pandemic at local, regional, and national levels. Results: Analysing patterns of four RSV seasons prior to and three seasons after the SARS-CoV-2 pandemic, we found that the paediatric RSV hospitalisation wave of the 2023/2024 season was weaker and less pronounced than expected. When we compared detailed local, regional, and national surveillance data of other viral infections, we found that paediatric RSV hospitalisation waves were anticyclical to SARS-CoV-2 infection waves, but not to rhinovirus or influenza waves in the general population. Discussion: Our data suggests that concomitant SARS-CoV-2 infection waves in the general population may disrupt infection chains of RSV in children and, thus, decrease RSV-associated hospitalisation. This factor should be taken into account when assessing the effects of the upcoming RSV prophylaxis in the future.

## 1. Introduction

Infections with human Respiratory Syncytial Viruses (RSVs) are a major cause for hospitalisation in children worldwide [1,2,3,4]. Only recently, we showed that RSV is also an important cause for respiratory illness, morbidity, and even mortality in elderly patients [5,6]. Our systematic observation since 2016 revealed ever-increasing RSV hospitalisation waves in children in South-eastern Germany (Eastern Bavaria) following a bi-annual pattern, where a high-prevalence season is always followed by a lower-prevalence season [6]. Until 2023, this pattern was only disrupted by the severe acute respiratory syndrome coronavirus 2 (SARS-CoV-2) pandemic, when a high-prevalence RSV season was expected for 2020 but was completely omitted, most likely due to non-pharmacological interventions (NPIs) aimed against coronavirus disease 2019 (COVID-19). This biannual pattern of RSV waves was not only described in our region, but also in other countries in northern Europe [7] as well as in Alberta, Canada [8]. In 2021, RSV hospitalisations in children in Eastern Bavaria reached pre-pandemic numbers predicted for a low-prevalence season [6], followed in the 2022/2023 season by the largest wave of RSV infections and hospitalisations ever recorded [9]. According to the bi-annual pattern of RSV infections, we anticipated a strong lower-prevalence season in 2023/2024, exceeding the 2021/2022 season. However, that did not occur.

Thus, we explored potential explanations for the deviation of this latest RSV hospitalisation wave in children from those we had documented thoroughly in our hospital since 2016. We analysed data from paediatric hospitals in Eastern Bavaria and surveillance data on acute respiratory infections on local, regional, and national levels based on different assessment methods to test whether other viral infections may interfere with paediatric RSV hospitalisation waves.

## 2. Materials and Methods

### 2.1. Local RSV Hospitalisation Data

Since 2016, we have thoroughly assessed hospitalisations for RSV in children in the University Children’s Hospital, Regensburg (KUNO), which is the only paediatric hospital in Regensburg and the only paediatric university hospital in Eastern Bavaria. During the general respiratory infection season (usually starting in November and lasting until March), we tested every patient who was hospitalised with symptoms of respiratory infection routinely by RT-PCR for RSV, Influenza A and B, and since the season of 2020/2021, also for SARS-CoV-2, using the commercial triplex assay Xpert™ Xpress FLU/RSV (2017–11/2020) on the GeneExpert™ platform (Cepheid) and since 12/2020, with the quadruplex Xpert™ Xpress CoV-2/Flu/RSV (Cepheid Inc., USA) [10,11].

In addition, hospitalisations for RSV were assessed at the following children’s hospitals in Eastern Bavaria (Figure 1): In Passau, located 130 km east of Regensburg, data on all RSV hospitalisations in children were provided for seasons since the end of 2018. In Altötting, 130 km south of Regensburg, data was provided for the season 2023/2024. RSV infections in both hospitals were diagnosed by RT-PCR using the same tests (Cepheid Inc., Sunnyvale, CA, USA) as Regensburg. In Weiden, 85 km north of Regensburg, data of RSV hospitalisations in children were provided for the season 2023/2024, diagnosed using the Savanna RVP4 (RVP4, QuidelOrtho Diagnostics, San Diego, CA, USA) test [12]. In addition, RSV hospitalisation data were provided by the children’s hospital Hof, 180 km north of Regensburg, diagnosed using the Panther Fusion SARS-CoV-2/FluA/B/RSV assay (Hologic Inc., Marlborough, MA, USA) [13].

### 2.2. Regional Bavarian RSV and SARS-CoV-2 Surveillance Data

Bavarian sentinel data from the Bavarian Influenza + Corona Sentinel (BIS-C), Figure 1 on acute respiratory infections collected in the primary care setting are reported by the Landesamt für Gesundheit und Lebensmittelsicherheit (LGL) based on 209 medical offices providing data in 2023/2024 [14,15]. Sewage water monitoring (Figure 1) for SARS-CoV-2 was established in Passau, Weiden, Hof, and Altöttng in 2022 and in Regensburg in 2023 [16,17].

### 2.3. National German RSV and SARS-CoV-2 Surveillance Data

The German national sentinel surveillance by the Robert Koch Institute (RKI) (Figure 1) collects data on acute respiratory infections from about 700 medical offices all over Germany, representing 1% of primary care centres [18], and these data are reported weekly [19,20]. In addition, hospitalisations due to respiratory infections are reported via the Severe Acute Respiratory Infection (SARI) Sentinel system, which covers about 6% of hospitalised patients in Germany. The hospitalisation index is calculated via the sentinel patient admission in relation to national patient admission and the general population (Figure 1) [21,22].

Furthermore, a national sewage water monitoring (Figure 1) for SARS-CoV-2 was established in 2022, and unbiased data on SARS-CoV-2 load in a number of regions and cities became available and publicly accessible [17].

### 2.4. Statistical Analysis

We analysed the data with Microsoft Excel (version 2016) and SPSS Statistics (version 28). For statistical evaluation, differences between two groups were assessed with Mann–Whitney U testing. When more than two groups were compared, Kruskal–Wallis testing was applied. We conducted the study in accordance with the Declaration of Helsinki. The retrospective analysis was approved by the local ethics committee (# 22-2973-104, 20 June 2022, and # 24-3764-104, 15 May 2024).

As all those hospitals are very close to each other and have different sizes, we normalised the hospitalisations, rather than comparing absolute numbers. Unfortunately, a census of babies and toddlers is not available for the catchment areas of the analysed hospitals but is based on other political structures (counties/districts). Therefore, we had to revert to the number of listed beds calculated by the Bavarian Ministry of Health on a formula that is mainly based on the number of children in the area, but also on other structural factors [23]. The start and end of an RSV season in relation to hospitalisation were defined retrospectively when all data of a season had been collected as previously described: The start of the RSV hospital season was defined as the date when 3% of all cases of that season were hospitalised [6]. The end of the season was reached when 97% of all cases of that season had been hospitalised. These dates were separately determined for each season.

## 3. Results

A total of 1665 children were hospitalised at the University Children’s Hospital Regensburg with a median age of 8 months (0–213 months) between autumn 2016 and spring 2024 (Figure 2).

The latest (lower prevalence) RSV season in Regensburg started in December 2023 and lasted until the start of April 2024. Compared to previous lower-prevalence RSV seasons, the 2023/2024 season was longer than expected (131 days compared to a mean duration of lower-prevalence seasons of 118 days (*p* = 0.39)), and the number of children hospitalised was slightly lower than in the last lower-prevalence season 2022/2023 (219 vs. 224, *p* = 0.32). In those 219 children, a coinfection with SARS-CoV-2 was detected in 9 patients (4.1%) and in 12 patients (5.5%) with influenza. We neither observed a significant difference in age distribution nor length of inpatient treatment between the current and previous RSV seasons (Table 1, Appendix A).

The 2023/2024 season showed two peculiarities in Regensburg compared to seasons since 2016: It did not follow the pattern of ever-increasing waves of previous lower-prevalence seasons (2021/2022, 2019/2020, and 2017/2018, respectively). Furthermore, this season was the first and only season that did not show a single peak in hospitalisations but instead, two lower peaks: The number of children hospitalised due to RSV decreased rather abruptly at the end of 2023, and a second low peak occurred only in February of 2024 (Figure 2). Interestingly, the same peculiarities of RSV hospitalisations were also observed for this season on the German national level (Figure 3a) [20,24].

As we had noticed a timely association between the abrupt halt of the RSV wave in December 2023 and a massive increase in sick leaves in our hospital staff due to SARS-CoV-2 at the same time, we sought to investigate if those events might be connected to each other. Thus, we first analysed publicly available national sentinel surveillance data provided by the RKI (Figure 3b): Indeed, RSV hospitalisation waves never occurred when SARS-CoV-2 detection rates peaked in the general population (Figure 3a,b) [20,21]. In contrast to SARS-CoV-2, neither the occurrence of rhinovirus (Figure 3c) nor influenza (Figure 3d) waves interfered with RSV hospitalisation waves in the national German sentinel surveillance data.

National sewage water monitoring for SARS-CoV-2 established in 2022 confirmed sentinel surveillance data and further strengthened the observation of a reciprocal association between SARS-CoV-2 and RSV waves in children (Figure 4): The SARS-CoV-2 wave, which we suspected to have stopped RSV infection chains in December of 2023, was the strongest SARS-CoV-2 wave ever recorded in Germany according to unbiased national sewage water monitoring data.

Next, we investigated if the same patterns in the occurrence of SARS-CoV-2 and RSV waves in children could be confirmed on the regional and local level, where timely associations could be investigated in more detail for the 2023/2024 season. We observed that the decrease in RSV-related hospitalisations in December 2023 occurred two weeks after the peak of SARS-CoV-2 in the Bavarian sentinel data and in the national sewage water monitoring was reached (Appendix A).

Finally, we could confirm our hypothesis in the datasets available from four other paediatric hospitals from Eastern Bavaria (Figure 5): Local sewage water monitoring was also available in Passau, Hof, Weiden, and Altötting for the 2023/2024 season. In Hof, Weiden, and Altötting, patterns of paediatric hospitalisations for RSV and SARS-CoV-2 detection in the general population were very similar to those in Regensburg (Figure 5a–c,e). In Passau, where 822 cases of RSV hospitalisations between autumn of 2018 and spring of 2024 had been documented, we found almost identical RSV waves as in Regensburg between 2018 and 2023 (Appendix A), but not for the season of 2023/2024: While the RSV season in Regensburg deviated from the expected pattern as described above, RSV hospitalisations in Passau showed a slightly higher peak compared to the 2021/2022 season (and in the total number of hospitalised children), exactly as predicted and expected (Figure 5d, Appendix A). Intriguingly, and against the national trend and data from Regensburg, Weiden, Hof, and Altötting, no exceptional SARS-CoV-2 wave was observed in Passau in December 2023 (Figure 5d). Thus, the data from Passau further strengthens the hypothesis that the concomitant occurrence of a strong SARS-CoV-2 wave in the winter of 2023 was an important factor in reducing RSV hospitalisation in children, and that relationship is not by chance.

## 4. Discussion

The 2023/2024 RSV hospitalisation wave in Regensburg and Germany showed a considerable deviation from expected patterns compared to previous RSV seasons since 2016. First, we observed two small peaks instead of one high peak in hospitalisation; second, it did not follow the pattern of ever-increasing waves of lower-prevalence seasons observed previously. These peculiarities were not seen before. To explore these differences, we analysed epidemiological data on viral infections on the national, regional, and local levels. We identified anti-cyclic patterns between RSV hospitalisations and SARS-CoV-2 but not rhinovirus or influenza waves. In the presence of an exceptional SARS-CoV-2 wave in the winter of 2023, RSV hospitalisation in children was diminished, as documented in four different regions in Eastern Bavaria, and it was unaffected when such an exceptional SARS-CoV-2 wave was absent, as observed in Passau.

Our study comprised data from five paediatric centres in Bavaria, for which we were able to analyse comprehensive datasets of RSV hospitalisations since 2016 in Regensburg, 2018 in Passau, and 2023/2024 in Hof, Weiden, and Altötting, respectively. RSV diagnoses were based on RT-PCR in all locations, and the indication for hospitalisation for RSV was very similar in all hospitals. Thus, neither RSV diagnostics nor rules for hospitalisation of children with RSV infections differed largely between hospitals, nor were they changed over time. Sentinel and sewage monitoring data were collected with the same methodology in all locations, as described previously [14,16,17,18,20]. A systematic difference exists between sentinel and sewage monitoring data. While the first measurement depends on the clinical presentation of patients, the sewage data monitoring is unbiased and, thus, virus levels detected with sewage monitoring are generally higher and truly population-based. Of note, sewage water data systematically excludes individuals wearing diapers.

Patterns of RSV from both Regensburg and Passau were very similar for all seasons until 2023/2024 (Appendix A). Thus, a general bias in collecting data in different locations seems very unlikely. In Regensburg, Hof, Weiden, Altötting, and on the national level, the SARS-CoV-2 infection wave in the general population in December of 2023 was the strongest ever observed since the first occurrence of that virus, but this exceptional SARS-CoV-2 wave was not observed in Passau.

If these associations are not coincidental but causal, the SARS-CoV-2 variant present in 2023 could have had specific features leading to protection from RSV infection (or a more severe RSV infection). This potential explanation could not be supported by experimental data from the literature. While interferon gamma release due to some virus infections may protect from further virus infections at the same time or shortly after [25,26], this has neither been described for SARS-CoV-2, in general, nor the current strain specifically [27]. In addition, coinfection with both viruses might have led to a shift in virus transmission and epidemiology. Infections with SARS-CoV-2 may inhibit RSV replication and vice versa [28,29]. A mathematical model exploring these interactions predicted the highest interaction for SARS-CoV-2 and RSV compared to SARS-CoV-2 and influenza or rhinovirus, which is in line with our finding [30]. Alternatively, the SARS-CoV-2 virus variant could have been exceptionally infectious, causing many infections and, at the same time, leading to large numbers of sick leaves and, thus, reducing the potential of RSV to spread due to a diminished number of socially active susceptible contacts. This is exactly what had been observed for the COVID-19 wave in the winter of 2023. Youth and middle-aged adults may play a central role in RSV transmission since, unlike elderly and children, they are normally not seriously affected by RSV infections or even asymptomatic, and, thus, can continue spreading RSV. An infection with SARS-CoV-2 may lead to a change in social behaviour and, therefore, to reduced virus transmission. Currently, little data exist on RSV epidemiology in this age group. High detection rates in the surveillance data and sewage water monitoring suggest relevant transmissions of RSV and SARS-CoV-2 also in this age group [14,15,16]. Moreover, a study in Italy detected frequent RSV infections in community-dwelling adults [31]. However, if young adults are seriously sick with SARS-CoV-2, they may neither be able to acquire RSV nor to spread it.

A limitation of our study is that we were not able to track the individual transmission routes of our RSV patients; therefore, a direct causal relation to the SARS–CoV-2 waves cannot be proven. Although not proven biologically, the interaction of SARS-CoV-2 and RSV in epidemiological terms seems to be the most probable explanation for the atypical RSV wave 2023/2024. Moreover, distinct local sentinel and sewage data were only available for the season 2023/2024. Therefore, we had to revert to a variety of measurements, such as the national surveillance data, for the comparison to other seasons, which may have led to less conclusive comparisons than if all measurements had been in place since the COVID-19 pandemic. As comprehensive surveillance data for RSV in the outpatient setting on the local level is lacking, we had to compare hospital cases with outpatient surveillance data, which may have led to a bias. However, RSV hospitalisations are directly related to outpatient cases, making it a good proxy, with an estimated ratio of 1:100 outpatient cases for each hospitalisation and a documented delay of 2 weeks between peaks in waves the outpatient setting in the community and hospitalisation peaks [20]. In addition, using different datasets and data sources allowed for a descriptive statistical approach only. However, surveillance data reported by the European Centre for Disease Prevention and Control (ECDC) show a similar pattern in other countries in central and northern Europe, e.g., Denmark and Norway [7] which emphasises our findings. Interestingly, other authors also observed a significant decrease in RSV-related hospitalisations in children for the 2023/2024 season [32]. Authors from Spain and France reported that the decrease in RSV hospitalisations in 2023/2024 is caused by the introduction of Nirsevimab. This antibody against RSV was first applied to newborns and children below 1 year of age, in general, in the autumn of 2023 in both countries and in Germany in the autumn of 2024, and is highly effective and significantly reduces RSV hospitalisations in infants [32]. However, two aspects have to be considered in the assessment of effects attributed to RSV prophylaxis in the future: (1) In Germany and some other northern countries, a biannual pattern of RSV seasons is well established, as we (and others) have shown previously [6,7,8]. In addition, as presented here, the association of SARS-CoV-2 waves has to be considered if they overlap with RSV seasons. Not to do so may over- or underestimate the impact of RSV prophylaxis, depending on the seasons that are compared.

## 5. Conclusions

The paediatric RSV hospitalisation wave in the 2023/2024 season was smaller than expected and showed a disrupted pattern not observed before. Based on data from five different hospitals and assessments of infection waves by other viruses, this may be best explained by a concomitant occurrence of a SARS-CoV-2 wave in the winter of 2023. We postulate an association between SARS-CoV-2 on RSV epidemiology, presumably through a disruption of transmission chains in the general population. Data from a location where no strong SARS-CoV-2 wave was observed and consequently, RSV hospitalisations did not decrease, support this hypothesis. When RSV hospitalisation rates and effects of antibody prophylaxis are assessed in the future, potential associations with SARS-CoV-2 need to be considered.

## Figures and Tables

**Figure 1 viruses-17-01584-f001:**
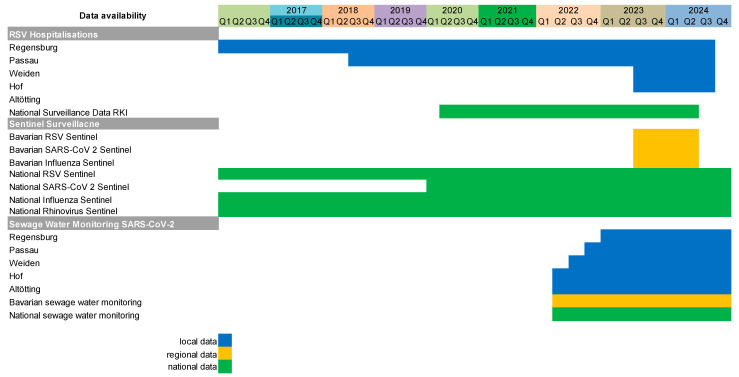
Data availability for analysis on local, regional (Bavarian), and national (German) levels.

**Figure 2 viruses-17-01584-f002:**
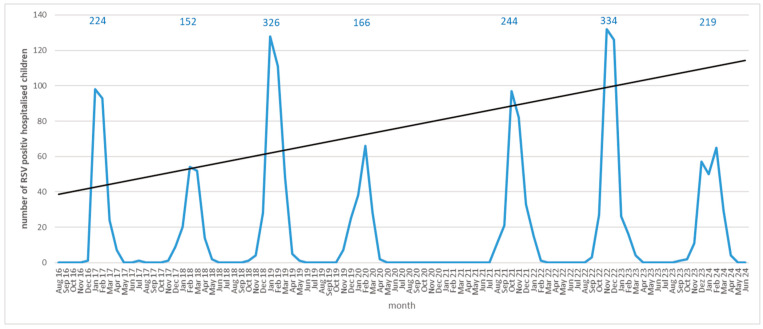
Number of children hospitalised with RSV infection per month (Regensburg). Black line: expected peak of the RSV hospitalizations in 2023/2024 calculated via the Excel forecast.linear function based on low-prevalence seasons 2017/2018, 2019/2020, and 2021/2022. (For syntax, see Appendix A).

**Figure 3 viruses-17-01584-f003:**
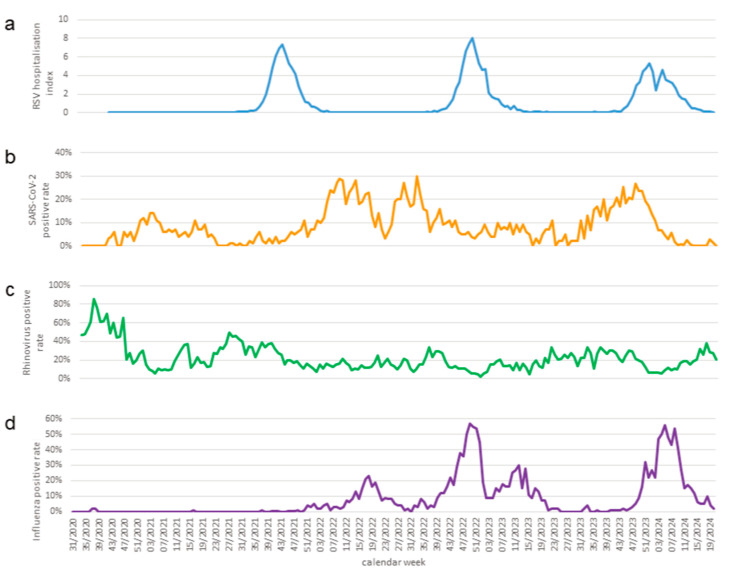
(**a**): National data for RSV-related hospitalisation in children [20,21]. (**b**): Positive rates for SARS-CoV-2, national sentinel surveillance [20]. (**c**): Positive rates for rhinovirus, national sentinel surveillance [20]. (**d**): Positive rates for influenza, national sentinel surveillance [20].

**Figure 4 viruses-17-01584-f004:**
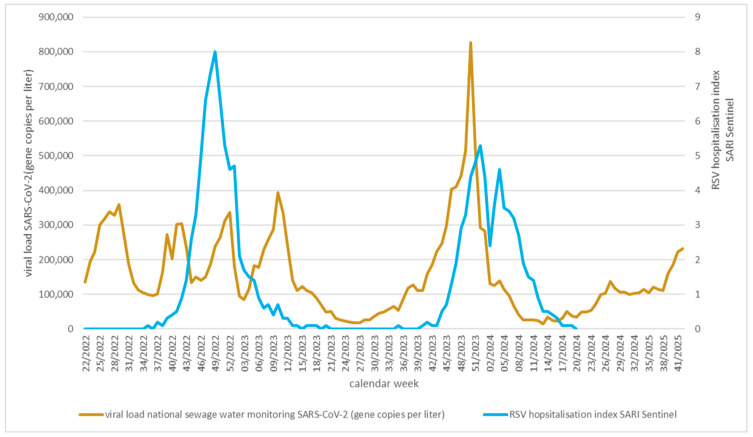
National SARS-CoV-2 sewage water monitoring [17] and RSV hospitalisation index (national sentinel) [20] for 2022/23 and 2023/24 seasons.

**Figure 5 viruses-17-01584-f005:**
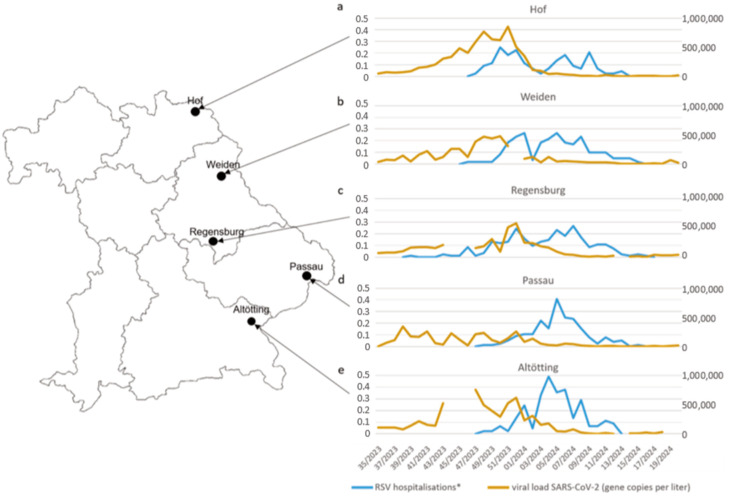
RSV hospitalisations * and local sewage water monitoring [17]. (**a**) Hof. (**b**) Weiden. (**c**) Regensburg. (**d**) Passau. (**e**) Altötting. * RSV Hospitalisations per week/listed beds [23] by calendar week.

**Table 1 viruses-17-01584-t001:** Age distribution and duration of inpatient stay of children hospitalised with RSV infections in Regensburg.

	2016/2017	2017/2018	2018/2019	2019/2020	2020/2021	2021/2022	2022/2023	2023/2024
n	224	152	326	166	0	244	334	219
Median age (months)	4	8	7	11		10	7	9
IQR	13.5	17	21	21		26	24.25	23
Median duration of inpatient treatment (days)	4	5	4	4		4	4	4
IQR	3	2	2	2		3	3	3
Inpatient treatment at least 7–10 days (absolute numbers and %)	27 (12.1)	26 (17.1)	33 (10.1)	16 (9.6)		33 (13.5)	37 (11.1)	28 (12.8)
Inpatient treatment at least 10 days (absolute numbers and %)	16 (7.1)	7 (4.6)	25 (7.7)	12 (7.2)		20 (8.2)	23 (6.9)	16 (7.3)

Grey collums: high prevalance season, white collums: low prevalence seasons.

## Data Availability

The raw data supporting the conclusions of this article will be made available by the authors on request.

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
