# Peer review of "Viruses2025, 17(12), 1584;https://doi.org/10.3390/v17121584"

_viruses, 2025, doi:10.3390/v17121584_

Round 1

Reviewer 1 Report

Comments and Suggestions for Authors

The article “Timely association of RSV hospitalisation waves in children with the detection of SARS-CoV-2 in the general population in Eastern Bavaria” describes the peculiarities of the 2023/2024 RSV season in this geographical region, which observed an unusual double peak.

The data based on screening (triple) tests carried out on children admitted to five hospitals in the region, as well as data on viral load from regional wastewater, national RSV and SARS-CoV-2 surveillance by the RKI and the Bavarian LGL, provide a robust basis for presenting and comparing the courses of RSV epidemics from 2016 to 2023. In this respect, the methodology and results are appropriate and reliable.

The discussion regarding the possible causes for this particular course of the 2023/24 RSV season remains speculative, of course. Assuming non-pharmaceutical measures such as social distancing measures due to the simultaneous SARS-CoV-2 outbreak is a simple and understandable explanation, which already led to impressive changes in the epidemiology of RSV infections in the 2020/2021 season.

More interesting is the possibility of co-infections influencing the course of the disease. I would like to see more discussion on this topic, as there are various studies (animal experiment data, case series, etc., see below) on co-infections with viral pathogens in childhood respiratory infections.

Literature, for example:

Carstens G, Kozanli E, Bulsink K, McDonald SA, Elahi M, de Bakker J, Schipper M, van Gageldonk-Lafeber R, van den Hof S, van Hoek AJ, Eggink D. Co-infection dynamics of SARS-CoV-2 and respiratory viruses in the 2022/2023 respiratory season in the Netherlands. J Infect. 2025 May;90(5):106474. doi: 10.1016/j.jinf.2025.106474. Epub 2025 Mar 21. PMID: 40122246.

Giannattasio A, Maglione M, D'Anna C, Muzzica S, Angrisani F, Acierno S, Perrella A, Tipo V. Silent RSV in infants with SARS-CoV-2 infection: A case series. Pediatr Pulmonol. 2021 Sep;56(9):3044-3046. doi: 10.1002/ppul.25465. Epub 2021 May 25. PMID: 34033702; PMCID: PMC8242430.

Morris DR, Qu Y, Thomason KS, de Mello AH, Preble R, Menachery VD, Casola A, Garofalo RP. The impact of RSV/SARS-CoV-2 co-infection on clinical disease and viral replication: insights from a BALB/c mouse model. bioRxiv [Preprint]. 2023 May 24:2023.05.24.542043. doi: 10.1101/2023.05.24.542043. PMID: 37292863; PMCID: PMC10245946.

Another point that should be addressed here is: is it possible to determine the proportion of vital co-infections (RSV, influenza, SARS-CoV-2) among hospitalized patients in the five hospitals? This could provide clues as to protective or disease-aggravating influences.

Furthermore, if available, a more detailed position should be taken on international data on the course of the 2023/2024 RSV season and the possible influence of viral co-infections on the course of the RSV season.

Author Response

1. Summary

We thank the reviewer for the helpful comments on the manuscript which we have edited to address the concerns.

2. Questions for General Evaluation

Reviewer’s Evaluation

Response and Revisions

Does the introduction provide sufficient background and include all relevant references?

Yes

Is the research design appropriate?

Yes

Are the methods adequately described?

Yes

Are the results clearly presented?

Yes

Are the conclusions supported by the results?

Are all figures and tables clear and well-presented?

Can be improved

Yes

3. Point-by-point response to Comments and Suggestions for Authors

Comments 1:

The discussion regarding the possible causes for this particular course of the 2023/24 RSV season remains speculative, of course. Assuming non-pharmaceutical measures such as social distancing measures due to the simultaneous SARS-CoV-2 outbreak is a simple and understandable explanation, which already led to impressive changes in the epidemiology of RSV infections in the 2020/2021 season.

More interesting is the possibility of co-infections influencing the course of the disease. I would like to see more discussion on this topic, as there are various studies (animal experiment data, case series, etc., see below) on co-infections with viral pathogens in childhood respiratory infections.

Response 1: Thank you for pointing this out. Indeed, co-infections with other viruses are common in children with RSV infections. A recently published mathematic model predicted a strong interaction of cocirculating RSV and SARS-CoV-2 infections and a weaker interaction of SARS-CoV-2 and rhinovirus or influenza.

(Pinky L, Dobrovolny HM. Epidemiological Consequences of Viral Interference: A Mathematical Modeling Study of Two Interacting Viruses. Front Microbiol. 2022 Mar 11;13:830423. doi: 10.3389/fmicb.2022.830423 . PMID: 35369460 ; PMCID: PMC8966706.)

We added this to our discussion section:

“In addition coinfection with both viruses might have led to a shift in virus transmission and epidemiology. Infections with SARS-CoV-2 may inhibit RSV replication and vice versa [28,29]. A mathematic model exploring these interactions predicted the highest interaction for SARS-CoV-2 and RSV compared to SARS-CoV-2 and influenza or rhinovirus which is in line with our finding [30]. line 255-259

Comment 2:

Another point that should be addressed here is: is it possible to determine the proportion of vital co-infections (RSV, influenza, SARS-CoV-2) among hospitalized patients in the five hospitals? This could provide clues as to protective or disease-aggravating influences.

Response 2:

Thank you for addressing this important point. Unfortunately, we are only able to provide data on coinfections for the hospital in Regensburg. Overall, the rate of coinfections was low, at 4.1 % for co-infections with RSV and SARS-CoV-2 and 5.5 % for co-infections with RSV and influenza in the 2023/2024 season in Regensburg.

We added this in the results section.

“In those 219 children a coinfection with SARS-CoV-2 was detected in 9 patients (4.1 %) and in 12 patients (5.5 %) with influenza” line 147-148.

Comments 3:

Furthermore, if available, a more detailed position should be taken on international data on the course of the 2023/2024 RSV season and the possible influence of viral co-infections on the course of the RSV season.

Response 3:

Thank you for this important comment. Indeed, surveillance data from 2023/2024 by the ECDC showed RSV and SARS-CoV-2 seasons similar to ours for example in Denmark and Norway. RSV epidemiology is different in countries in southern Europe and it is speculated that the warmer climate is responsible for that.

We added this to the discussion section:

”As comprehensive surveillance data for RSV in the outpatient setting on the local level is lacking, we had to compare hospital cases with outpatient surveillance data which may have led to a bias. However, RSV hospitalisations are directly related to outpatient cases, making it a good proxy, with an estimated ratio of 1:5-10 outpatient cases for each hospitalisations and a documented delay of 2 weeks between peaks in waves the outpatient setting in the community and hospitalisation peaks [21]. In addition, using different datasets and data sources allowed for a descriptive statistical approach only. However surveillance data reported by the European Centre for Disease Prevention and Control (ECDC) show a similar pattern in other countries in central and northern Europe e.g. Denmark and Norway[7] which emphasizes our findings.” line 281-290

Reviewer 2 Report

Comments and Suggestions for Authors

This research employs multicenter surveillance data to propose that RSV transmission may be suppressed due to SARS-CoV-2 circulation. This is certainly a novel phenomenon. Indeed, the correlational data is weak. Overall, the report is more of an intriguing ecological observation than it is of scientific proof of an underlying mechanistic conclusion.  

  1. The article posits that waves of SARS-CoV-2 RSV may possibly disrupt RSV circulation, but there is a lack of mechanistic data and individual-level tracing; there is no recourse but to rely on temporal “inverse associations.”
  2. Other explanations, such as epidemiological school closures, changes in societal behavior, or even competition from other infections, are still very much on the table. To pin the findings exclusively on SARS-CoV-2 is grossly reductionist.  
  3. The authors present Passau as a “counterexample”, but do not examine other geographic or analytic contextual variables, such as local social behavior or surveillance coverage.  
  4. The intermarginal RSV epidemic, and the question that it raises, is in consideration of whether the interval between the two peaks is the result of immunity gaps, or age structure within the population.  
  5. The research is concerned primarily with hospital cases, while the biases introduced by the disregard for outpatients or infections of the community may skew the perception of the magnitude of the RSV epidemic. The “synchrony”that is observed with the peaks of SARS-CoV-2 and falls of RSV may be pure coincidence.
  6. Discussion of Nirsevimab captures SDA deficiency - its resolution lacks coherence.
  7. Figures 2, 3, 5 are unnecessary duplicates - their content could be combined and made more concise.

Author Response

1. Summary

We thank the reviewer for the helpful comments on the manuscript which we have edited to address the concerns.

2. Questions for General Evaluation

Reviewer’s Evaluation

Response and Revisions

Does the introduction provide sufficient background and include all relevant references?

Must be improved

Is the research design appropriate?

Must be improved

Are the methods adequately described?

Must be improved

Are the results clearly presented?

Must be improved

Are the conclusions supported by the results?

Are all figures and tables clear and well-presented?

Must be improved

Must be improved

3. Point-by-point response to Comments and Suggestions for Authors

Comment 1:

The article posits that waves of SARS-CoV-2 RSV may possibly disrupt RSV circulation, but there is a lack of mechanistic data and individual-level tracing; there is no recourse but to rely on temporal “inverse associations.”

Response 1:

Indeed what we describe is an epidemiological inverse correlation of SARS-CoV-2 detection in the general population and RSV hospitalisations in children. We already stated in the discussion: „A limitation of our study is that we were not able to track the individual transmission routes of our RSV patients, therefore a direct causal relation to the SARS–CoV-2 waves cannot be proven.“ In order to emphasis this point we have modified the discussion section.

“Although not proven biologically, the interaction of SARS-CoV-2 and RSV in epidemiological terms seems to be the most probable explanation for the atypical RSV wave 2023/2024.” line 275-277

On the other hand, a recently published mathematic model predicted a strong interaction of cocirculating RSV and SARS-CoV-2 infections and a weaker interaction of SARS-CoV-2 and Rhinovirus or Influenza virus.

(Pinky L, Dobrovolny HM. Epidemiological Consequences of Viral Interference: A Mathematical Modeling Study of Two Interacting Viruses. Front Microbiol. 2022 Mar 11;13:830423. doi: 10.3389/fmicb.2022.830423 . PMID: 35369460 ; PMCID: PMC8966706.)

We added this to our discussion section.

“A mathematic model exploring these interactions predicted the highest interaction for SARS-CoV-2 and RSV compared to SARS-CoV-2 and influenza or Rhinovirus which is in line with our finding [30]“ line 257 - 259

Comment 2:

Other explanations, such as epidemiological school closures, changes in societal behavior, or even competition from other infections, are still very much on the table. To pin the findings exclusively on SARS-CoV-2 is grossly reductionist.  

Response 2:

As shown in the peak of the SARS-CoV-2 pandemic 2020/2021, school closures have the potential to disrupt infection chains and reduce virus transmission. However, there were no school closures reported for the analyzed regions in 2023/2024 other than Christmas vacations, which are a recurring event, happening every year and not affecting RSV waves in any previous season we documented since 2016. The potential changes in social behavior especially in young adults, due to a SARS-CoV-2 infection, is exactly what we postulate as one potential factor to reduce RSV transmission. We adjusted our discussion section to emphasize this point.

“An infection with SARS-CoV-2 may lead to a change in the social behavior and therefore, to a reduced virus transmission” line 266 - 267

In addition, we agree that a direct competition from other viruses remains another possible explanation for a change in virus transmission. This has already been addressed in the publication. The reported national surveillance data show no correlation with rhinovirus or influenza. Obviously, we cannot report on other viruses that are not detected in the surveillance data. In order to make it clear that there are other possible explanation we adjusted the discussion section.

„Although not proven biologically, the interaction of SARS-CoV-2 and RSV in epidemiological terms seems to be the most probable explanation for the atypical RSV wave 2023/2024“ line 275 - 277

Comment 3:

The authors present Passau as a “counterexample”, but do not examine other geographic or analytic contextual variables, such as local social behavior or surveillance coverage.  

Response 3:

We agree that we were not able to analyze potential differences in the social behavior between Passau and the other reported regions. However, as all the reported regions are in the same close geographic area (Eastern Bavaria), inhabited by the same ethnic population with very similar social „Bavarian“ behavior, relevant differences in social behavior seem extremely unlikely. RSV hospitalisations in Passau from season 2018/2019 until 2022/2023 followed the same pattern as in Regensburg, which is shown in Supplement Figure 3.

RSV testing was performed with the same tests and indications in all locations. The sewage water monitoring, which is performed by the Bavarian public health authorities is also performed with the same method in all regions. Therefore, a systematic bias in the surveillance or the RSV hospiatlisation data can be ruled out.

Comment 4:

The intermarginal RSV epidemic, and the question that it raises, is in consideration of whether the interval between the two peaks is the result of immunity gaps, or age structure within the population.  

Response 4:

In contrast to the post SARS-CoV-2 pandemic season there is no reason why immunity gaps in the general population should have occurred in 2023/2024 and no data exists, that would suggest such a gap now. Even the first post pandemic RSV seasons in Regensburg, in which an immunity gap after months of the pandemic could have occurred and influenced RSV hospitalisations, did show one and not two peaks. In addition, no significant change in age structure was detected (which could have suggested such an immunity gap from the COVID-19 pandemic). This is shown in Figure 2 and Supplement Figure 1.

Comment 5:

The research is concerned primarily with hospital cases, while the biases introduced by the disregard for outpatients or infections of the community may skew the perception of the magnitude of the RSV epidemic. The “synchrony”that is observed with the peaks of SARS-CoV-2 and falls of RSV may be pure coincidence.

Response 5:

The analysis was indeed performed with hospital cases. There are two reasons why we relied on hospitalised RSV cases: (1) Comprehensive surveillance data for RSV in the outpatient setting is lacking, especially on the local level. (2) RSV hospitalisations are directly related to outpatient cases, making it a good proxy, with an estimated ratio of 1:100 outpatient cases for each hospitalisation and a documented delay of 2 weeks between peaks in waves the outpatient setting in the community and hospitalisation peaks as described previously (Koch-Institut, Robert; Influenza, A. Wochenberichte. Available online: https://influenza.rki.de/Wochenberichte.aspx (accessed on 22 October 2024). Thus, we do not agree with the reviewer that the comparison of surveillance and hospitalisation data may lead to a bias. We added this point in the discussion section.  

“As comprehensive surveillance data for RSV in the outpatient setting on the local level is lacking, we had to compare hospital cases with outpatient surveillance data which may have led to a bias. However, RSV hospitalisations are directly related to outpatient cases, making it a good proxy, with an estimated ratio of 1:100 outpatient cases for each hospitalisation and a documented delay of 2 weeks between peaks in waves the outpatient setting in the community and hospitalisation peaks” line 281 - 286

Comment 6:

Discussion of Nirsevimab captures SDA deficiency - its resolution lacks coherence.

Response 6:

Thank you for this comment. We have to make it clear that the prophylaxis with Nirvesimab is highly effective. Our conclusion does not doubt this effect. We emphasized this in the discussion section. “This antibody (…) is highly effective and significantly reduces RSV hospitalisations in infants” line 294-297

However, as shown in our own data and in other countries a biannual pattern of RSV hospitalisations and the peculiarity of season 2023/2024 are independent factors that should be taken into account when assessing the effect of a prophylaxis especially in these regions.

“However surveillance data reported by the European Centre for Disease Prevention and Control (ECDC) show a similar pattern in other countries in central and northern Europe e.g. Denmark and Norway[7] which emphasizes our findings.“ line 288-290

Comment 7:

Figures 2, 3, 5 are unnecessary duplicates - their content could be combined and made more concise.

Response 7:

In this point we disagree with the reviewer. These figures may present similar data but not the same. The have been carefully designed to address specific aspects and make these visible and understandable to the interested reader. Thus, we would suggest not to combine or dismiss them.

Figure 2 presents RSV hospitalisation data in Regensburg from season 2016/2017 onwards. This graph highlights both peculiarities of season 2023/2024 (two peaks and less hospitalisations than the last (low) prevenance season)

Figure 3 presents national surveillance data showing the inverse correlation of SARS-CoV-2 detections and RSV hospitalisations in the sentinel data. In addition, the presentation of Rhinovirus and Influenza detections in the sentinel systems shows that for those two viruses, an inverse correlation is not present.

Figure 5 presents distinct local data of five regions for the season 2023/2024, with a different SARS-CoV-2 and RSV hospitalisation wave in Passau compared to all other regions.

We believe that the presentation of the findings in the different levels, which indeed show similar results, strengthens these findings and therefore is important for the manuscript.

Reviewer 3 Report

Comments and Suggestions for Authors

The main criticism of the manuscript submitted to the Viruses is that the authors did not indicate what is new and relevant in their paper abd why the manuscript may be interesting to the reader

Moreover, the authors suggest that "paediatric RSV hospitalisation wave in the 2023/2024 season showed an unexpected pattern not observed before", but it is difficult to draw such conclusions from the results presented in the manuscript

Some minor issues should also be resolved

1) Whether the Fig. 1 should be designated as Table? Since it is a table?

2) Please, add subchapters for Chapter 2.

3) line 18: excel --> Excel?

4) Captions to Figures should be located under the Figures

5) Fig. 2: positive --> positive?

6) Please, discuss in the lines after Table 1, why there were no a single case detected in 2020/2021 season

7) Fig. 4: sewag --> sewage?

Author Response

1. Summary

We thank the reviewer for the helpful comments on the manuscript which we have edited to address the concerns.

2. Questions for General Evaluation

Reviewer’s Evaluation

Response and Revisions

Does the introduction provide sufficient background and include all relevant references?

Must be improved

Is the research design appropriate?

Must be improved

Are the methods adequately described?

Must be improved

Are the results clearly presented?

Must be improved

Are the conclusions supported by the results?

Are all figures and tables clear and well-presented?

Must be improved

Must be improved

3. Point-by-point response to Comments and Suggestions for Authors

Comments 1:

The main criticism of the manuscript submitted to the Viruses is that the authors did not indicate what is new and relevant in their paper abd why the manuscript may be interesting to the reader

Response 1:

We are very sorry that the reviewer does neither find the paper interesting nor that we could make him appreciate new and relevant aspects our work adds to the field of RSV epidemiology in children. As the reviewer does not give specific reasons for his opinion, it is not possible to respond to that comment in a structured manner. However, keeping this comment in mind, we adjusted the writing of the manuscript in several sections to make novelty and relevance more obvious to more interested readers.

Some minor issues should also be resolved

1) Whether the Fig. 1 should be designated as Table? Since it is a table?

Indeed the data could have also been reported via a table. However the way it is presented we believe that it is rather a figure than a table.

2) Please, add subchapters for Chapter 2.

In chapter 2 contains 4 subchapters

Local RSV hospitalisation data (line 76)

Regional Bavarian RSV and SARS-CoV-2 surveillance data (line 96)

National German RSV and SARS-CoV-2 surveillance data (line 103)

Statistical analysis (line 15)

3) line 18: excel --> Excel?

We changed excel to Excel

4) Captions to Figures should be located under the Figures

Thank you for pointing this out. We located the captions under the figures.

5) Fig. 2: positive --> positive?

We changed Positive to positive in Supplemental Figure 2

6) Please, discuss in the lines after Table 1, why there were no a single case detected in 2020/2021 season

The RSV season 2020/2021 was completely omitted, therefore no patient was hospitalised in our hospital in this season. This is stated in the manuscript „Until 2023, this pattern was only disrupted by the severe acute respiratory syndrome coronavirus 2 (SARS-CoV-2) pandemic, when a high prevalence RSV season was expected for 2020 but was completely omitted, most likely due to non-pharmacological interventions (NPIs) aimed against coronavirus disease 2019 (COVID-19)” line 56-60

7) Fig. 4: sewag --> sewage?

The spelling error has been corrected

Reviewer 4 Report

Comments and Suggestions for Authors

The study describes the incidence of RSV in a German population across various seasons.
The authors report the biseasonal distribution of RSV, which was interrupted by the SARS-CoV-2 pandemic, while the distribution of rhinovirus and influenza A-B was not impacted by the COVID pandemic.
Unfortunately, the study cannot be transferred to other regions as this biseasonal distribution has not been documented.
It is unclear whether the epidemiology of influenza A-B and rhinovirus has been evaluated in hospitalized patients.

Author Response

1. Summary

We thank the reviewer for the helpful comments on the manuscript which we have edited to address the concerns.

2. Questions for General Evaluation

Reviewer’s Evaluation

Response and Revisions

Does the introduction provide sufficient background and include all relevant references?

Yes

Is the research design appropriate?

Can be improved

Are the methods adequately described?

Can be improved

Are the results clearly presented?

Can be improved

Are the conclusions supported by the results?

Are all figures and tables clear and well-presented?

Can be improved

Must be improved

3. Point-by-point response to Comments and Suggestions for Authors

Comment 1:

The study describes the incidence of RSV in a German population across various seasons.

The authors report the biseasonal distribution of RSV, which was interrupted by the SARS-CoV-2 pandemic, while the distribution of rhinovirus and influenza A-B was not impacted by the COVID pandemic.

Unfortunately, the study cannot be transferred to other regions as this biseasonal distribution has not been documented.

It is unclear whether the epidemiology of influenza A-B and rhinovirus has been evaluated in hospitalized patients.

Response 1:

Thank you for this thoughtful comment. Indeed, the biannual pattern of RSV waves is not present everywhere. It is described in many areas with cool climate in the northern hemisphere such as Denmark and Estonia, as shown by ECDC surveillance data. The same pattern has also been described in Alberta, Canada in the pre SARS-CoV-2 period. (Griffiths C, Drews SJ, Marchant DJ. 2017. Respiratory syncytial virus: infection, detection, and new options for prevention and treatment. Clin Microbiol Rev 30:277–319.

https://doi.org/10.1128/CMR.00010-16).

We now added this data to the introduction section. “This biannual pattern of RSV waves was not only described in our region, but also in other countries in northern Europe as well as in Alberta, Canada” line 60-62

The epidemiology of rhinovirus and influenza in hospitalised children has not been evaluated in the study.

This is due to different reasons. For rhinovirus no systematic screening at hospital admission exists so far in our or the other contributing centers, nor any other pediatric hospitals we are aware of. Therefore, we have no data to anaylze hospitalisations due to Rhinovirus.

Influenza regularly occurs at the same time as RSV, without disrupting the RSV wave in the past. This was already reported for our hospital (Kiefer A, Kabesch M, Ambrosch A. The Frequency of Hospitalizations for RSV and Influenza Among Children and Adults. Dtsch Arztebl Int. 2023 Aug 7;120(31-32):534-535. doi: 10.3238/arztebl.m2023.0111. PMID: 37721140; PMCID: PMC10534134) but is also apparent in the present surveillance data.

However, rhinovirus, influenza and RSV hospitalisations were evaluated in the national surveillance data which showed no association between Rhinovirus or Influenza with RSV. This is shown in Figure 3 and stated in the results “In contrast to SARS-CoV-2, neither the occurrence of rhinovirus (Figure 3c) nor influenza (Figure 3d) waves interfered with RSV hospitalisations waves in the national German sentinel surveillance data.” line 189-192

Reviewer 5 Report

Comments and Suggestions for Authors

This manuscript investigates the unusual bi-phasic RSV hospitalisation wave in children during the 2023/24 season in Eastern Bavaria. Using local hospital data, regional sentinel data, and national surveillance (including sewage water monitoring), the authors report an anti-cyclical relationship between RSV hospitalisations and SARS-CoV-2 infection waves, but not influenza or rhinovirus. They hypothesise that intense SARS-CoV-2 waves disrupt RSV transmission chains, thereby reducing RSV-related hospitalisations.

The topic is timely and relevant for pediatric infectious disease epidemiology, especially in the context of RSV prophylaxis implementation. However, the manuscript requires clarification of methodology, stronger statistical justification, and refinement of its discussion to avoid overinterpretation.

1)The manuscript suggests that SARS-CoV-2 reduced RSV hospitalisations. However, the data are observational and ecological. Please be more cautious in language  “association” is appropriate, while “impact” or “direct effect” may overstate the evidence.

2)The current statistical approach (mainly Mann-Whitney and Kruskal-Wallis) is descriptive. Given the strong claims, more robust time-series analysis (e.g., cross-correlation, interrupted time series, or regression modeling) would better test the anti-cyclical association

3)RSV hospitalisations were normalized by number of pediatric beds (p. 5 ). This is unusual and may not reflect true incidence rates. Could population denominators (children <5 years) be used instead? At least justify why bed capacity is the appropriate denominator.

4) Figures 3–5 are central but difficult to interpret:

  • Axes should be labeled more clearly (hospitalisations per 100k children, viral copies per liter, etc.).

  • Ensure consistent scales across panels for comparability.

  • Indicate the exact timing of SARS-CoV-2 peaks vs RSV troughs with vertical markers.

5)Similar disruptions of RSV by SARS-CoV-2 waves may have been reported elsewhere. Please expand the discussion with evidence from other countries . This will strengthen generalisability.

Minor Comments

  1. Abstract is clear but too long; consider condensing.

  2. The introduction repeats known RSV epidemiology — streamline for conciseness.

  3. Define all abbreviations at first use (e.g., SARI, BIS-C).

  4. Please proofread for grammar (e.g., “ther abruptly” on p. 8 should read “rather abruptly”

Author Response

1. Summary

We thank the reviewer for the helpful comments on the manuscript which we have edited to address the concerns.

2. Questions for General Evaluation

Reviewer’s Evaluation

Response and Revisions

Does the introduction provide sufficient background and include all relevant references?

Can be improved

Is the research design appropriate?

Yes

Are the methods adequately described?

Can be improved

Are the results clearly presented?

Can be improved

Are the conclusions supported by the results?

Are all figures and tables clear and well-presented?

Can be improved

Can be improved

3. Point-by-point response to Comments and Suggestions for Authors

Comment 1:

The manuscript suggests that SARS-CoV-2 reduced RSV hospitalisations. However, the data are observational and ecological. Please be more cautious in language  “association” is appropriate, while “impact” or “direct effect” may overstate the evidence.

Response 1:

Thank you for this comment. We rephrased several statements.

“In addition, as presented here, association of SARS-CoV-2 waves have to be considered if they overlap with RSV seasons” line 301-302

“We postulate an association between SARS-CoV-2 on RSV epidemiology presumably through a disruption of transmission chains in the general population” line 310–311

When RSV hospitalisation rates and effects of antibody prophylaxis are assessed in the future, potential associations with SARS-CoV-2 need to be considered. line 314-316

Comment 2:

The current statistical approach (mainly Mann-Whitney and Kruskal-Wallis) is descriptive. Given the strong claims, more robust time-series analysis (e.g., cross-correlation, interrupted time series, or regression modeling) would better test the anti-cyclical association

Response 2:

We agree that the statistical approach is mainly descriptive and we stated that in the paper. We never claimed that is more than a first observational report. More advanced statistical analysis would have been much preferred if the available data would allow for that. Unfortunately, data are collected from different datasets (sentinel surveillance, sewage water monitoring, hospitalised patients) and on different levels (national, regional, local). Furthermore, the season 2023/2024 is the first and only season which presented with concomitant SARS-CoV-2 and RSV waves. Therefore, a statistical robust correlation is not possible based on one observation.

We added this to the discussion section „In addition, using different datasets and data sources allowed for a descriptive statistical approach only” line 286-288

Comment 3:

RSV hospitalizations were normalized by number of pediatric beds (p. 5). This is unusual and may not reflect true incidence rates. Could population denominators (children <5 years) be used instead? At least justify why bed capacity is the appropriate denominator.

Response 3:

Thank you for reading the manuscript so thoroughly that you noticed this. Indeed, we had quite some discussion in the team how to approach the problem: The Hospitals we analyzed are very close to each other and have different sizes trsanslating to (available) beds. The number of beds are/have been calculated by the ministry of health on a formula that is based on the number of children in the area, but not only on that. The basis of the calculation is not available. A census of babies and toddlers is not available for the catchment areas of the analyzed hospitals but based on other political structures (counties/districts). Thus, to normalize for the available beds (which are based on political quota and the reality of patient care in the region) is the closest we could get and the best available surrogate marker for the served population.

We added this in the methods section:

“As all those hospitals are very close to each other and have different sizes we normalized the hospitalisations, rather than comparing absolute numbers. Unfortunately, a census of babies and toddlers is not available for the catchment areas of the analyzed hospitals but based on other political structures (counties/districts). Therefore, we had to revert to the number of listed beds calculated by the Bavarian ministry of health on a formula that is mainly based on the number of children in the area, but also on other structural factors.” line 122 - 128

Comment 4:

Figures 3–5 are central but difficult to interpret:

Axes should be labeled more clearly (hospitalisations per 100k children, viral copies per liter, etc.).

Ensure consistent scales across panels for comparability.

Indicate the exact timing of SARS-CoV-2 peaks vs RSV troughs with vertical markers.

Response 4:

In figure 3, we specified that the first graph represents the hospitalisation index, which is not reported in percent. In addition, the three viruses which are presented follow a different epidemiology with difference especially in peaks. Therefor a change to consistent scales in this figure would avert the presentation of the SARS-CoV-2 waves.

Regarding figure 2 we changed the title to reflect the order in which the data are presented

For both figures we need to make it clear how the SARI hospitalisation index is calculated as it is not self-explaining why this is not presented as a percentage. The SARI hospitalisation index is calculated via the sentinel patient admissions in in relation to national patient admission and the general population as described by Tolksdorf et al (Tolksdorf, K.; Haas, W.; Schuler, E.; Wieler, L.H.; Schilling, J.; Hamouda, O.; Diercke, M.; Buda, S. ICD-10 based syndromic surveillance enables robust estimation of burden of severe COVID-19 requiring hospitalization and intensive care treatment. medRxiv 2022, 2022.02.11.22269594, doi:10.1101/2022.02.11.22269594). We added this to the method section. “In addition hospitalisations due to respiratory infections are reported via the Severe acute respiratory infections (SARI) Sentinel system, which covers about 6 % of hospitalised patients in Germany. The hospitalisation index is calculated via the sentinel patient admission in relation to national patient admission and the general population” line 107-110

Comment 5:

Similar disruptions of RSV by SARS-CoV-2 waves may have been reported elsewhere. Please expand the discussion with evidence from other countries . This will strengthen generalizability.

Response 5:

Thank you for addressing this point. Similar disruptions are observable in other countries in central and northern Europe in the surveillance data of the ECDC. We added this to the discussion section. “However surveillance data reported by the European Centre for Disease Prevention and Control (ECDC) show a similar pattern in other countries in central and northern Europe e.g. Denmark and Norway[7] which emphasizes our findings” line 288-290

Minor Comments:

Abstract is clear but too long; consider condensing.

          We shortened the abstract

The introduction repeats known RSV epidemiology — streamline for conciseness.

We believe that the description of the regular RSV epidemiology is crucial to understand the significance of the prescribed peculiarities in the season 2023/2024. Therefore, we would be reluctant to change the introduction.

Define all abbreviations at first use (e.g., SARI, BIS-C).

          We changed the manuscript in order to explain all abbreviations at first use

Please proofread for grammar (e.g., “ther abruptly” on p. 8 should read “rather abruptly”

          The manuscript has been proofread for grammar

Round 2

Reviewer 2 Report

Comments and Suggestions for Authors

none

Author Response

We are glad that the reviewer agrees with the revised version of the manuscript.

Reviewer 3 Report

Comments and Suggestions for Authors

The authors made the changes, which were recommended by the reviewers, but the two comments are still relevant and were not changes in the submitted manuscript 

- they did not indicate what is new and relevant in their paper and why the manuscript may be interesting to the reader

- this conclusion is not still obvious for the readers "paediatric RSV hospitalisation wave in the 2023/2024 season showed an unexpected pattern not observed before"

Author Response

1. Summary

We thank the reviewer for the helpful comments on the manuscript which we have edited to address the concerns.

2. Questions for General Evaluation

Reviewer’s Evaluation

Response and Revisions

Does the introduction provide sufficient background and include all relevant references?

Must be improved

Is the research design appropriate?

Must be improved

Are the methods adequately described?

Must be improved

Are the results clearly presented?

Must be improved

Are the conclusions supported by the results?

Are all figures and tables clear and well-presented?

Must be improved

Must be improved

3. Point-by-point response to Comments and Suggestions for Authors

Comment 1:

The authors made the changes, which were recommended by the reviewers, but the two comments are still relevant and were not changes in the submitted manuscript 

- they did not indicate what is new and relevant in their paper and why the manuscript may be interesting to the reader

- this conclusion is not still obvious for the readers "paediatric RSV hospitalisation wave in the 2023/2024 season showed an unexpected pattern not observed before"

Response 1:

The main finding of the study is that the RSV hospitalistation wave in 2023/2024 was different to the previous seasons. First, we observed two small instead of one high peak in hospitalisation, second, it did not follow the pattern of ever-increasing waves of lower-prevalence seasons observed previously . These peculiarities were not seen before, neither in our hospital nor in the other reported hospitals. The analysis of the surveillance data on different levels showed a timely association with a high detection rate of SARS-CoV-2 in the general population. Although not proven biologically, the interaction of SARS-CoV-2 and RSV in epidemiological terms seems to be the most probable explanation for the atypical RSV wave 2023/2024. This interaction was not described before. In our opinion these findings are new and relevant and therefore interesting to readers and justify the conclusion “The paediatric RSV hospitalisation wave in the 2023/2024 season showed an unexpected pattern not observed before.”

In order to make these findings more understandable we adjusted the conclusion section as follows.

The paediatric RSV hospitalisation wave in the 2023/2024 season was smaller than expected and showed a disrupted pattern not observed before.

Reviewer 5 Report

Comments and Suggestions for Authors

thank you 

Author Response

(The authors gave the same response as above.)
